# Activating de novo mutations in *NFE2L2* encoding NRF2 cause a multisystem disorder

Peter Huppke[1,2], Susann Weissbach[1], Joseph A. Church[3], Rhonda Schnur[4], Martina Krusen[5], Steffi Dreha-Kulaczewski[1], W. Nikolaus Kühn-Velten[6], Annika Wolf[1], Brenda Huppke[1], Francisca Millan[7], Amber Begtrup[7], Fatima Almusafri[8], Holger Thiele[9], Janine Altmüller[9,10], Peter Nürnberg[9,11,12], Michael Müller[2,13] & Jutta Gärtner[1]

Transcription factor NRF2, encoded by *NFE2L2*, is the master regulator of defense against stress in mammalian cells. Somatic mutations of *NFE2L2* leading to NRF2 accumulation promote cell survival and drug resistance in cancer cells. Here we show that the same mutations as inborn de novo mutations cause an early onset multisystem disorder with failure to thrive, immunodeficiency and neurological symptoms. NRF2 accumulation leads to widespread misregulation of gene expression and an imbalance in cytosolic redox balance. The unique combination of white matter lesions, hypohomocysteinaemia and increased G-6-P-dehydrogenase activity will facilitate early diagnosis and therapeutic intervention of this novel disorder.

[1] Department of Pediatrics and Adolescent Medicine, Division of Pediatric Neurology, University Medical Center Göttingen, 37075 Göttingen, Germany. [2] Center for Nanoscale Microscopy and Molecular Physiology of the Brain (CNMPB), 37075 Göttingen, Germany. [3] Divison of Clinical Immunology and Allergy, Childrens Hospital Los Angeles, and Keck School of Medicine University of Southern California, Los Angeles, CA 90027, USA. [4] Division of Genetics, Cooper University Health Care, Cooper Medical School of Rowan University 3, Camden, NJ 08103, USA. [5] Lebenszentrum Königsborn Fachklinik für Kinderneurologie und Sozialpädiatrie mit Sozialpädiatrischem Zentrum, 59425 Unna, Germany. [6] Medical Laboratory Bremen, 28357 Bremen, Germany. [7] GeneDx, Gaithersburg, MD 20877, USA. [8] Department of Pediatrics, Clinical and Metabolic Genetics, Hamad Medical Corporation, 3050 Doha, Qatar. [9] Cologne Center for Genomics (CCG), University of Cologne, Cologne 50931, Germany. [10] Institute of Human Genetics, Universitätsklinik Köln, 50931 Cologne, Germany. [11] Center for Molecular Medicine Cologne (CMMC), University of Cologne, 50931 Cologne, Germany. [12] Cologne Excellence Cluster on Cellular Stress Responses in Aging-Associated Diseases (CECAD), University of Cologne, 50931 Cologne, Germany. [13] Zentrum Physiologie und Pathophysiologie, Institut für Neuro- und Sinnesphysiologie, Georg-August-Universität Göttingen, Universitätsmedizin, 37075 Göttingen, Germany. Peter Huppke and Susann Weissbach contributed equally to this work. Correspondence and requests for materials should be addressed to P.H. (email: phuppke@med.uni-goettingen.de)

The survival of cells relies on an immediate reaction to different insults such as oxidative stress, hypoxia, toxins and infections. The broad spectrum of genes involved in this defense mechanism share an antioxidant response element (ARE) in their regulator region that is recognized by nuclear factor-erythroid 2-related factor 2 (NRF2). NRF2 belongs to the Cap 'n' Collar (Cnc) family of basic leucine zipper transcription factors and regulates the expression of more than 200 genes[1, 2]. Under stressed conditions, NRF2 translocates into the nucleus where it accumulates, forming heterodimers with small Maf (musculo aponeurotic fibrosarcoma) proteins which then bind to AREs thereby activating the expression of the respective target genes[3]. Under non-stressed conditions NRF2 is rapidly inactivated to avoid unnecessary gene transcription. To achieve downregulation NRF2 is bound in the cytoplasm by a homodimer of the Kelch-like ECH-associated protein 1 (KEAP1), a cysteine-rich protein anchored to the actin cytoskeleton[4]. KEAP1 assembles with the Cul3 protein to form a Cullin–RING E3 ligase complex leading to ubiquitination of NRF2 thereby targeting it for degradation by the 26S proteasome[5]. Due to its rapid degradation the half-life of NRF2 is only 20 min under non-stressed conditions[6]. KEAP1 binds to NRF2 in the amino-terminal Nrf2 ECH homology 2 (Neh2) domain, one of seven functional Neh domains identified so far[7]. Two key amino acid sequences within Neh2, ETGE and DLG, facilitate the binding of the two KEAP1 molecules[8]. A hinge and latch mechanism has been proposed in which the ETGE motif, binding with high-affinity, acts as a hinge and the weaker DLG motif as the latch[8]. Under stressed conditions reactive cysteines within KEAP1 are modified by electrophiles and oxidants leading to a conformational change of KEAP1 and release from the DLG binding sequence thereby preventing ubiquitination and degradation of NRF2. This increases the half-life of NRF2 allowing it to transactivate stress response genes.

In this article we describe four patients with a multisystem disorder characterized by failure to thrive, immunodeficiency and neurological symptoms who carry inborn de novo missense mutations in *NFE2L2*. The mutations affect the binding sites of KEAP1 leading to accumulation of NRF2 and consecutive increased expression of genes regulated by NRF2.

## Results

**Identification of activating *NFE2L2* mutations in four patients.** Patient 1 is the second born child of non-consanguineous parents originating from India. He presented to the Pediatric Neurology department in Göttingen, Germany, at an age of 6 years with suspected multiple sclerosis. Since his first year of life, multiple hospital admissions for poor weight gain, growth retardation and recurrent lung and skin infections had occurred. Also evident from an early age were a generalized weakness, fatigue and inability to walk long distances. Physical examination revealed a cooperative yet very shy, underweight and growth retarded 6-year-old boy (weight 16 kg (< third percentile), length 112 cm (third percentile) body mass index: 12.6 (< third percentile) occipitofrontal circumference (OFC) 50 cm (third percentile)). He displayed no focal neurological signs or dysmorphic features but was easily fatigued on exertion. Furthermore, intelligence testing

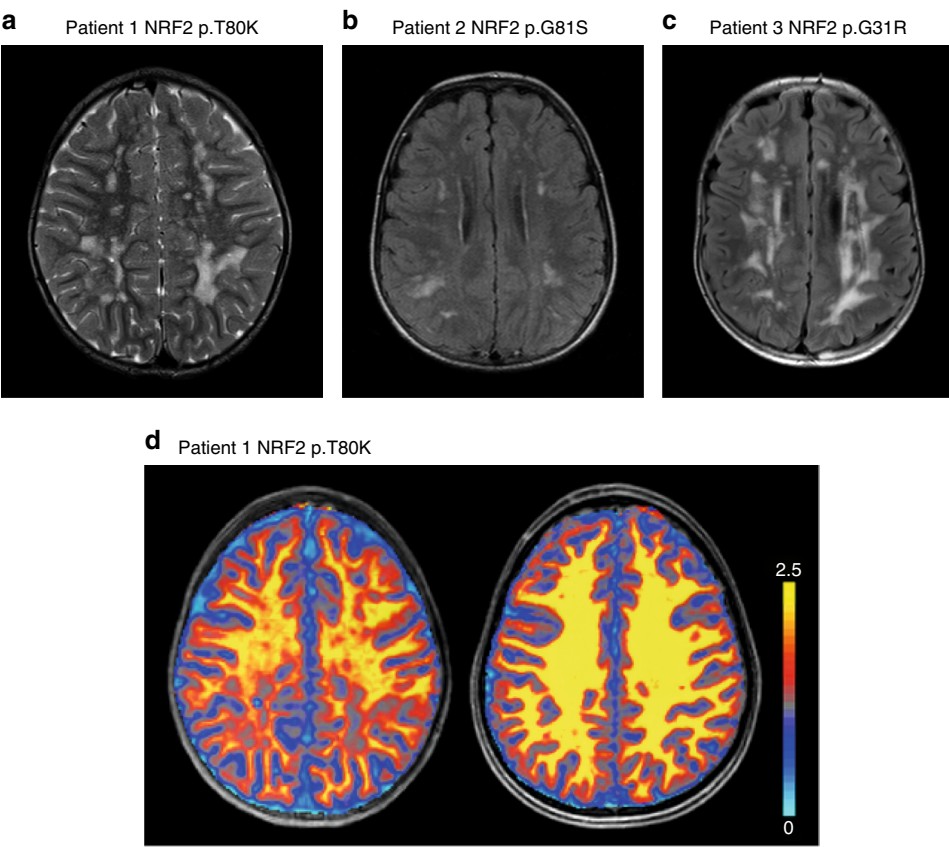

**Fig. 1** Activating mutations in NRF2 are associated with supratentorial white matter signal changes on MRI. T2 weighted images (**a**, patient 1) or FLAIR (**b**, patient 2 and **c**, patient 3) from three patients showing multiple smaller single or larger confluent hyperintense lesions. **d** Colorcoded MT sat maps overlayed onto the corresponding T1-weighted image of patient 1 (left) and age- and gender-matched healthy control (right). Note the distinct reduction of MT sat within the white matter lesions most pronounced in the occipital regions displaying values close to gray matter. Color scale with the respective MT sat values on the right

revealed an IQ of 74. Cerebral magnetic resonance (MR) imaging at age 6.9 years demonstrated bilateral periventricular and sub-cortical white matter signal hyperintensities on T2 weighted images with sparing of infratentorial and spinal structures (Fig. 1a). No contrast enhancement was demonstrated and on serial studies over 3 years the lesions remained unchanged. Proton magnetic resonance spectroscopy (MRS) showed reduced levels of N-acetylaspartylglutamate and creatine and normal levels of lactate in gray and white matter. On magnetization transfer saturation (MT sat) maps, a quantitative MR parameter for evaluating myelination, the lesions showed a distinct myelin deficit (Fig. 1d). Laboratory studies showed signs of liver damage, reduced homocysteine (2.9 µmol/l, reference range: 5.5–16.2 µmol/l), and low cysteine (5.0 µmol/l, reference range for the age of the patient: 5–45 µmol/l) levels in blood. Analysis of energy metabolism showed mildly elevated lactate in blood (2.8 mmol/l, reference range: 0.5–2.2 mmol/l), and cerebrospinal fluid (CSF) (2.0 mmol/l, reference range 1.1–1.8 mmol/l).

Mendeliome patient-parent trio sequencing (mean coverage of 71–80×, Supplementary Table 1) after intensive filtering revealed only a few rare compound heterozygous, hemizygous and de novo variants (Supplementary Table 2a). Of these only one, the heterozygous de novo variant c.239C>A in *NFE2L2* (NM_006164.4), could be related to homocysteine metabolism. The variant was predicted to have a functional impact by 7 out of 10 in silico analysis tools and has not been listed in the Exome Aggregation Consortium (ExAC) population database (Supplementary Table 2b). At the protein level, the c.239C>A mutation leads to a p.T80K substitution in NRF2 at a location within the ETGE motif of the Neh2 domain, the hinge that facilitates the binding to KEAP1 (Fig. 2). With the help of GeneMatcher (https://genematcher.org/), three further patients with mutations in *NFE2L2* were detected[9] (Supplementary Table 3).

Patient 2, a 13-year-old Caucasian boy from New Jersey, USA, and his parents underwent whole exome sequencing analysis for an undiagnosed disorder characterized by recurrent infections of the sinuses, lung and skin, short stature with delayed bone age, feeding problems with recurrent choking and aspiration, severe failure to thrive, mild developmental delay, absence seizures, and chronic headaches (Table 1). He had silvery blonde hair and malar and plantar erythema. Borderline liver enlargement was present as well as a bicuspid aortic valve with thickening of the aortic cusps. He had an intermittent tremor and a hypernasal and hoarse speaking voice. Lab studies showed hypogammaglobulinemia and IgA deficiency with poor polysaccharide response. Cerebral magnetic resonance imaging (MRI) showed a white matter lesion pattern similar to that seen in patient 1 (Fig. 1b). He was found to carry a de novo heterozygous missense mutation, c.241G>A/p.G81S also affecting the ETGE motif of NRF2, adjacent to the change in patient 1 (Fig. 2).

Patient 3 originating from Los Angeles, CA, USA, had a history of combined immune deficiency, short stature, failure to thrive, mild developmental/speech delay and eczematous lesions of the face and neck. Exome sequencing revealed a heterozygous de novo missense mutation, p.G31R, affecting the DLG motif of NRF2, the region that acts as the latch in the binding mechanism with the KEAP1 homodimer (Table 1, Fig. 2). Like patient 1, he was also found to have a reduced blood homocysteine level. MRI showed a similar pattern to that seen in the other two patients (Fig. 1c).

Patient 4, a 20-month-old Qatari girl was born at term as part of a non-identical twin pair to non-consanguineous parents. Intrauterine growth retardation was present and birth weight was below the third percentile (1.7 kg). The patient showed mild

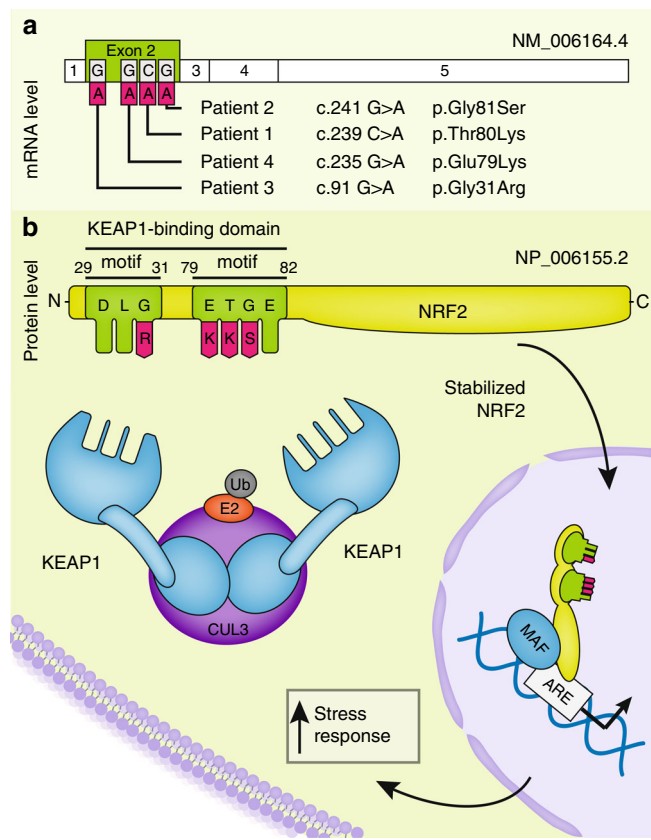

**Fig. 2** Location and functional consequences of the identified NRF2 variants. **a** Location of the mutations in the *NFE2L2* gene. All *NFE2L2* mutations that have been identified are heterozygous de novo missense mutations in exon 2. **b** Linear representation of the NRF2 polypeptide showing the detailed position of the mutations in the Neh2 domain. All mutations are located either in the DLG or the ETGE motif. These motifs are essential for binding two molecules KEAP1 in unstressed conditions leading to rapid degradation of NRF2. The mutations inhibit the binding of KEAP1 thereby increasing NRF2 levels in the absence of stress and consecutive chronic activation of stress response genes

dysmorphic features and developmental delay affecting especially fine motor skills. Common primary immunodeficiency with natural killer cell dysfunction was suspected due to recurrent severe chest infections. So far no MRI has been performed. Laboratory data are shown in Table 1. Like in patient 1 (not investigated in patients 2 and 3) cysteine in plasma was low (16.0 µmol/l, reference range for the age of the patient: 16–84 µmol/l). Exome sequencing revealed a de novo missense mutation c.235G>A /p.E79K affecting the ETGE motif from NRF2.

The four de novo NRF2 variants p.G31R, p.E79K, p.T80K, and p.G81S were each confirmed using Sanger sequencing (Supplementary Fig. 1). All are located in highly conserved residues of the ETGE and DLG motifs of the Neh2 domain (PhastCons, GERP, Supplementary Table 4a) and are predicted to have a functional impact by the majority of in silico analysis tools (Supplementary Table 4b).

In summary, exome sequencing revealed four apparently pathogenic variants in *NFE2L2* that are each associated with a very similar early onset clinical phenotype. The location of the mutations strongly suggests that all of them lead to impaired binding of KEAP1. Thus, our findings strongly indicate that all four patients suffer from the same, previously undescribed multisystem disorder.

**Table 1 Summary of clinical features of patients harboring *NFE2L2* mutations**

| Patients Variant | 1 c.239C > A; p.T80K | 2 c.241G > A; p.G81S | 3 c.91G > A; p.G31R | 4 c.235G > A p.E79K | Reference range |
|---|---|---|---|---|---|
| Inheritance | De novo | De novo | De novo | De novo | |
| Sex | Male | Male | Male | Female | |
| Age (years) | 9 | 13 | 14 | 1.8 | |
| Dystrophy | + | + | + | + | |
| Short stature | − | + | + | NA | |
| Delayed bone age | − | + | + | NA | |
| Muscle weakness | + | − | + | − | |
| Mild developmental delay | + | + | + | + | |
| Learning disability | + | + | + | NA | |
| Chronic headaches | + | + | − | NA | |
| Rec. lung infections | + | + | + | + | |
| Rec. skin infections | + | + | + | − | |
| Heart defects | − | Thickened bicuspid aortic valve | ASD, cardiomyopathy | ASD | |
| Homocysteine | ↓ (2.9 µmol/l) | ND | ↓ (1.6 µmol/l) | ↓ (3 µmol/l) | 5.5–16.2 µmol/l |
| Creatinine | ↓ (16.8 µmol/l) | ↓ (31 µmol/l) | ↓ (26.5 µmol/l) | ↓ (33 µmol/l) | 53–80 µmol/l |
| AST | ↑ (73 U/l) | ↔ (25 U/l) | ↑ (57 U/l)[a] | ↔ 21(U/l) | 26–55 U/l |
| ALT | ↑ (107 U/l) | ↑ (33 U/l) | ↑ (75 U/l)[a] | ↔(7 U/l) | 11–30 U/l |
| IGF1 | ↓ | ↓ | ↓ | ND | [b] |
| GSR | ↑ (19.7 U/g Hb) | ND | ND | ND | 5.0–11.0 U/g Hb |
| G6PD | ↑ (29.4 U/g Hb) | ND | ↑(14.4 U/g Hb) | ND | 7.2–10.5 U/g Hb |
| Immunoglobulin A | ↔ (138 mg/dl) | ↓(47 mg/dl) | ↓ (52 mg/dl) | ↓ (28 mg/dl) | (62–236 mg/dl) |
| Immunoglobulin G | ↔ (1110 mg/dl) | ↓ (541 mg/dl) | ↓ (494 mg/dl) | ↓ (319 mg/dl) | (698–1560 mg/dl) |
| Immunoglobulin M | ↔ (99 mg/dl) | ↓ (21 mg/dl) | ↓ (18 mg/dl) | ↔(130 mg/dl) | (31–179 mg/dl) |
| Switched Memory B-cells | ↓ | ND | ↓(<1/mcL) | ND | |
| Antibody response to Pneumovax[tm] | ↓ (positive in 1 of 6 serotypes) | ↓ (positive in 2 of 10 serotypes) | ↓ (positive in 2 of 23 serotypes) | NA | [c] |

NA not applicable, ND not done, GSR glutathione reductase, G6PD glucose-6-phosphate dehydrogenase
[a]AST/ALT intermittently elevated; normal at other times
[b]Laboratory and age specific normal range
[c]A response to Pneumovax[tm] is considered positive if it is >1.3 µg/ml antibody in at least 70% of the serotypes tested

**Effect of the *NFE2L2* mutations**. Each *NFE2L2* variant that was detected in the patients has previously been reported as a somatic cancer-related pathogenic change in several tumor species[10–16] (Supplementary Fig. 2). Somatic *NFE2L2* variants in the DLG and ETGE motif in cancer cells have been found to be gain of function mutations with an increased NRF2 stabilization due to impaired binding of KEAP1 resulting in an increased expression of several stress response genes[17]. In order to show that an identified de novo variant has a similar effect we analyzed human primary fibroblasts of patient 1 carrying the p.T80K NRF2 variant. First, we analyzed NRF2 expression by western blot analysis and found it significantly increased when compared to healthy control fibroblasts thereby demonstrating NRF2 accumulation due to reduced degradation (Fig. 3a and b, Supplementary Fig. 4a and b, Supplementary Table 5). Second, we performed quantitative real-time PCR (qRT–PCR) to investigate the effect of elevated NRF2 levels on the expression of genes that are known to be under control of NRF2. As expected, we observed an increased expression of genes involved in glutathione production and regeneration (*GCLM* (glutamate–cysteine ligase modifier), *GSR* (glutathione reductase), NADPH production (*G6PD, ME1* (malic enzyme 1)), thioredoxin production, regeneration and utilization (*TXNRD1* (Thioredoxin reductase 1), *PRDX1* (peroxiredoxin 1)), enzymes regulating iron sequestration (*HMOX1* (Heme oxygenase 1)), and drug excretion (*ABCC1* (ATP-binding cassette (ABC) subfamily C member 1))[18–20]. The strongest increase in expression was seen for the aldo-keto reductase (*AKR*) *1C1* and *AKR1B10* genes (Fig. 3c, Supplementary Table 5). Third, we analyzed the protein levels of KEAP1, G-6-P-dehydrogenase (G6PD), AKR1C1, and AKR1B10 demonstrating that the

enhanced transcription also leads to increased protein synthesis (Fig. 3a and b, Supplementary Fig. 4). Finally, we showed that the effect of NRF2 accumulation can be demonstrated not only in vitro but also in vivo by testing G6PD and GSR activity in erythrocytes of patient 1. Enzyme activity of G6PD was increased to 29.4 U/g Hb (reference range: 7.2–10.5 U/g Hb) and GSR activity to 19.7 U/g Hb (reference: 50–110 U/g Hb). The finding of elevated G6PD in erythrocytes was also seen in patient 3 (not analyzed in patient 2). In conclusion, biochemical analysis demonstrated that *NFE2L2* mutations have functional consequence leading to elevated NRF2 protein levels with consecutively increased expression of stress response genes in vitro and in vivo.

**Effect of NRF2 activation on cellular redox conditions**. As NRF2 induces antioxidant genes, we speculated that chronic activation might cause a more reducing resting state redox balance in the cytosol. We took advantage of the genetically encoded optical redox indicator roGFP1 (reduction oxidation sensitive green fluorescent protein 1) that is ratiometric by excitation and thus enables a quantitative assessment of subcellular redox conditions in living cells[21]. First, calibration of the response properties was performed by determination of the ratiometric responses corresponding to full oxidation (evoked by 10 mM $H_2O_2$, up to 5 min) and full reduction (evoked by 5 mM DTT, up to 5 min; Fig. 3d)[22]. On the basis of these data, the relative roGFP1 oxidation levels of cytosolic roGFP1 in fibroblasts as well as the corresponding redox potentials were calculated. For control fibroblasts this yielded an average

relative oxidation level of 52.9 ± 6.1 %, whereas for the fibroblasts of NRF2 patient 1 the relative oxidation level averaged only 20.6 ± 2.6 % (Fig. 3e, Supplementary Table 5). This corresponded to redox potentials of −287.5 ± 4.4 and −309.6 ± 2.1 mV, respectively, and confirmed a more reducing, i.e., 22 mV more negative,

resting state redox balance in the cytosol of NRF2 patient 1's fibroblasts.

**Treatment of chronic NRF2 activation in vitro and in vivo.** A search for possible compounds reported to reduce Nrf2 levels but

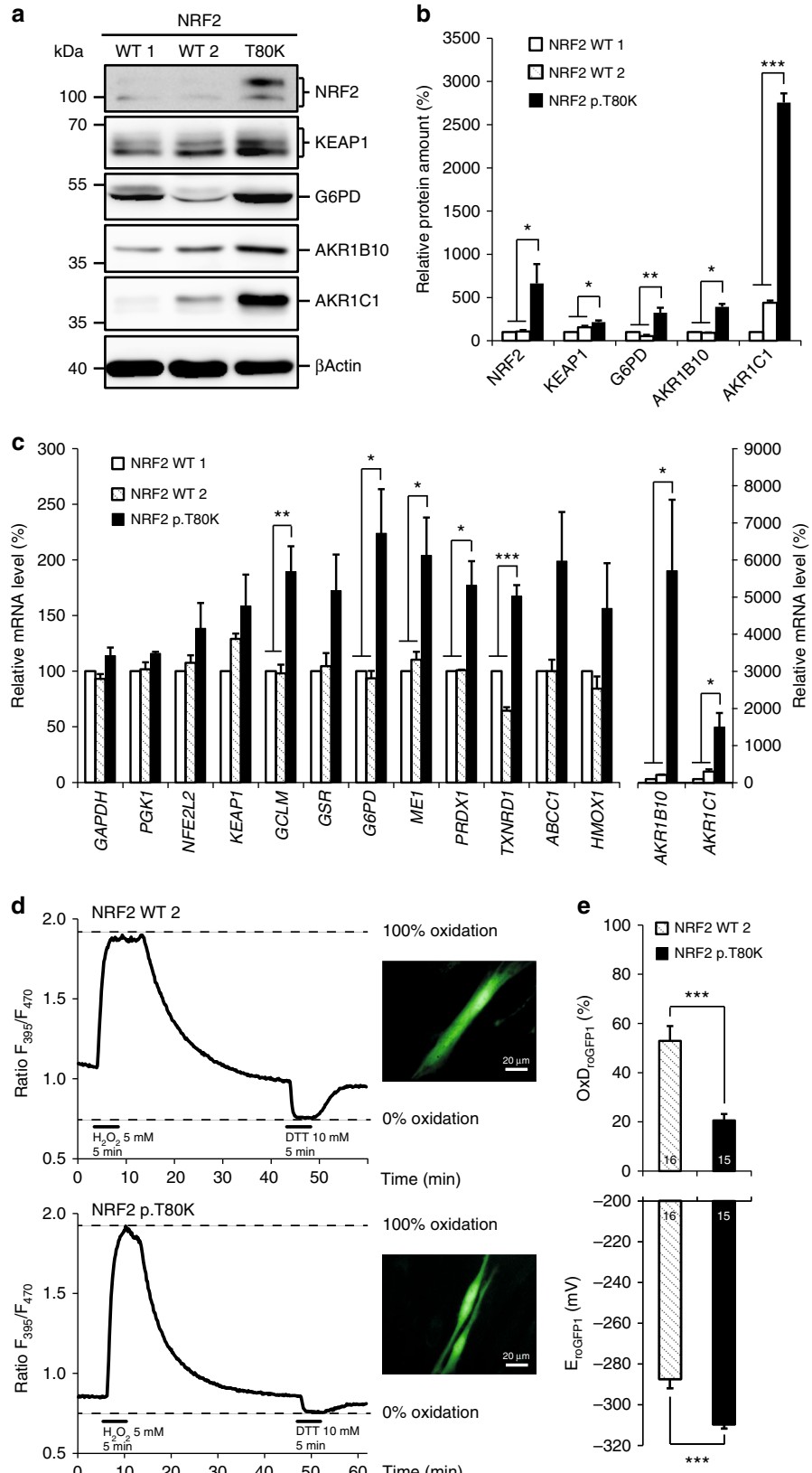

devoid of serious side effects when used in a clinical setting resulted in the identification of two candidates; ascorbic acid and luteolin[23]. While ascorbic acid is commonly used in clinical medicine, luteolin, a flavone that is meant to have anti-inflammatory and neuroprotective effects, is not commonly used in Western medicine so far. To test the effect of luteolin and ascorbic acid in vitro, fibroblasts from patient 1 were treated for 24 h with 50 μM luteolin or different concentrations of ascorbic acid. Subsequent western blotting showed that luteolin treatment reduced the NRF2 level up to 90% (Fig. 4a and b) while treatment with ascorbic acid was less effective and did not consistently lower NRF2 level to the same degree at the concentrations tested (Supplementary Fig. 5). qRT–PCR analysis of AKR1B10 and AKR1C1 gene expression, expression of both genes is upregulated by NRF2, in fibroblasts of patient 1 treated with 50 μM luteolin showed that downregulation of NRF2 resulted in significantly reduced expression of AKR1B10 and AKR1C1 (Fig. 4c).

Patient 1 was then treated with luteolin (50 mg once daily) and ascorbic acid (200 mg once daily). After 6 months of treatment previously elevated liver enzymes had normalized but no effect was seen on the homocysteine level. Furthermore, the mother reported a very positive development. Infections had been less frequent and less severe and the patient was able for the first time to carry his bag to school and attend the sport lessons. Moreover, overall school performance had improved.

## Discussion

In this article we describe four patients with a novel disorder caused by mutations in NFE2L2 that impair the binding of NRF2 by KEAP1. The patients do not have significant dysmorphic features, but display a similar phenotype with several prominent features including developmental delay, failure to thrive, immunodeficiency, leukoencephalopathy, and hypohomocysteinaemia.

Somatic missense amino acid substitutions affecting the DLG or ETGE motifs in the regulatory Neh2 domain of NRF2 are present in many types of cancer and have been found to be associated with a poor prognosis[17, 24]. Analysis of the COSMIC databank (cancer.sanger.ac.uk) revealed that the mutations detected in the four patients have been described previously in different kinds of cancer cells (Supplementary Fig. 2b). Shibata et al.[17] studied the effect of the p.T80R mutation within the ETGE motif, the residue affected in patient 1, in 293T cells and showed that it substantially reduced the ability to interact with KEAP1[17]. An NRF2 protein harboring a mutation in the DLG motif, as seen in patient 3, retained binding to KEAP1 but caused reduced NRF2 ubiquitination[17]. In 293T cells both mutations led to accumulation of NRF2 in the nucleus and subsequent increased expression of stress response genes. Correspondingly, we found elevated levels of NRF2 and increased expression of genes that are part of the NRF2 mediated stress response in fibroblasts of patient

1 (Fig. 3). The strongest increase in expression was seen for AKR1C1 and AKR1B10. This result corresponds well to the findings of MacLeod et al.[20] who knocked down KEAP1 in HaCaT keratinocytes and found that expression of AKR1C1 and AKR1B10 increased 12-and 16-fold.

All experiments on chronic NRF2 activation so far have been done in vitro. The patients described herein enabled us to analyze the in vivo effects of NRF2 upregulation. Enzyme activities of G6PD and GSR were determined in blood of patients 1 and 3 (Table 1). 3-fold elevation of G6PD activity and 2-fold of GSR activity clearly demonstrated an in vivo effect of the mutations.

One of the hallmarks of this novel disorder is hypohomocysteinaemia. Several disorders with high levels of homocysteine have been described causing generalized vascular damage and thromboembolic complications but so far no disorder is known that is associated with reduced levels. Hypohomocysteinaemia is most likely a direct effect of NRF2 activation as NRF2 positively regulates glutathione synthesis for which homocysteine serves as a precursor. Cysteine, which was also found reduced in the blood of patient 1 and 4 (not investigated in patient 2 and 3), is generally considered the limiting component of glutathione biosynthesis[25]. It has been found in HepG2 cells that as much as half of the cysteine used for glutathione biosynthesis is generated from homocysteine utilizing the transsulfuration pathway[26]. In the next step γ-glutamylcysteine is synthesized from cysteine and L-glutamate. This reaction is catalyzed by glutamate–cysteine ligase (GCL) and is the rate limiting step in glutathione synthesis. We found that the expression of GCL was increased almost 2-fold in the fibroblasts of patient 1 (Fig. 3). The final step of glutathione synthesis, the C-terminal addition of glycine, is catalyzed by glutathione synthetase, an enzyme also positively regulated by NRF2.

We speculated that the permanently increased level of NRF2 would affect the redox balance in the cytosol because NRF2 is physiologically upregulated as a response to oxidative stress. Accordingly, we found overexpression of enzymes necessary for the generation and regeneration of the two major antioxidants glutathione and thioredoxin. Determination of the redox conditions in NRF2 mutant fibroblasts of patient 1 using the optical redox indicator roGFP1 indeed confirmed a marked reducing shift of the cytosolic redox balance (Fig. 3e) which leads to increased reduction stress. Proteome reactivity profiling indicates that 890 human proteins are potentially sensitive to redox modulation resulting in either gain or loss of function[27]. Thus, potential cellular dysfunction by misregulated proteins will affect many pathways thereby further enhancing the negative effect of NFR2 accumulation.

In 2003, Wakabayashi et al.[28] reported on a mouse model carrying a Keap1-null mutation that led to constitutive chronic Nrf2 activation, as seen in the patients presented here. Size and

**Fig. 3** Increased stabilization and activation of mutant NRF2. **a** Representative western blot of endogenous level of NRF2, KEAP1, G6PD, AKR1B10 and AKR1C1 in protein lysates of human primary fibroblast cell lines from two controls (NRF2 WT 1, WT 2) and patient 1 with NRF2 p.T80K variant. Full blots are shown in Supplementary Fig. 6. **b** Quantitative analysis of western blot images illustrating the endogenous level of NRF2, KEAP1, G6PD, AKR1B10 and AKR1C1 relative normalized to ACTB and NRF2 WT 1. **c** qRT–PCR analysis of NFE2L2, KEAP1 and target gene expression in primary fibroblast cell lines from two controls (NRF2 WT 1, WT 2) and patient 1 with NRF2 p.T80K variant. AKR1B10 and AKR1C1 are visualized on a separated X axis due to the high range. Expression is normalized to that of ACTB. % of mRNA is equal to $2^{-\Delta\Delta CT}$ and normalized relative to NRF2 WT 1. Redox calibration confirms full functionality of roGFP1 as well as identical response ranges for NRF2 WT 2 and NRF2 p.T80K fibroblast cells. **d** Response range calibration of an exemplary NRF2 WT 2 and NRF2 p.T80K fibroblast cell performed as a continuous recording of the roGFP1 ratio $F_{395}/F_{470}$ within a ROI of cytoplasm of the cell, scale bar is 20 μM. Plotted traces represent full oxidation ($R_{ox}$, induced by 5 mM $H_2O_2$, 5 min) and full reduction ($R_{red}$, induced by 10 mM DTT, 5 min). After calibration the relative degrees of roGFP1 oxidation and corresponding roGFP1 redox potentials can be calculated. **e** Baseline redox conditions of NRF2 WT 2 and NRF2 p.T80K fibroblasts. Upper diagram shows the relative level of roGFP1 oxidation of NRF2 WT 2 and NRF2 p.T80K cells at rest (OxD$_{roGFP1}$, Eq. 1). Lower diagram represents corresponding steady-state roGFP1 redox potential (E$_{roGFP1}$, Eq. 2). **b**, **c** Data are given as means $\pm$ SEM, $n \geq 3$ independent experiments. Data were analyzed by one-way analysis of variance with multiple comparisons: *$p \leq 0.05$, **$p \leq 0.01$, ***$p \leq 0.001$. **e** Data are given as means $\pm$ SEM. Number of measured cells are given within the bar. Statistical differences were obtained with unpaired Welch's t-test: ***$p \leq 0.001$

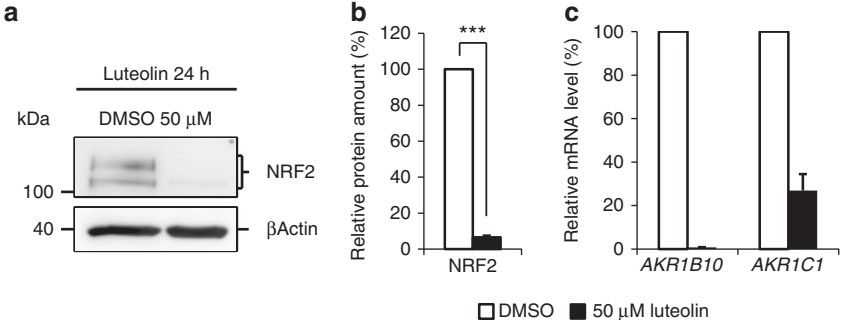

**Fig. 4** Downregulation of NRF2 induced by luteolin. **a** Representative western blot of endogenous level of NRF2 in whole protein lysates of treated human primary fibroblast of patient 1. NRF2 p.T80K mutant cells were exposed to 50 μM luteolin or DMSO for 24 h. Full blots are shown in Supplementary Fig. 7. **b** Quantitative analysis of western blot images illustrating the endogenous level of NRF2 relative normalized to ACTB and DMSO treated cells. Data are given as means ± SEM, $n = 3$ independent experiments. Statistical differences were obtained with unpaired Welch's $t$-test: ***$p \leq 0.001$. **c** qRT–PCR analysis of AKR1B10 and AKR1C1 gene expression in primary fibroblasts of patient 1 treated with 50 μM luteolin or DMSO for 24 h. Expression is normalized to that of GAPDH. % of mRNA is equal to $2^{-\Delta\Delta CT}$ and normalized relative to DMSO. Data are given as means ± SEM, $n = 2$ independent experiments

behavior of the $Keap1^{-/-}$ mutant mouse pups at birth were indistinguishable from those of wild-type littermates. Beginning around postnatal day 4 severe growth retardation was observed and all mutant mice had died by day P21 due to gradually progressive asthenia. Histological examination showed no morphological abnormalities except for hyperkeratosis in the esophagus and forestomach. Failure to thrive and growth retardation were also very prominent symptoms in the patients presented here. Endoscopy performed on patients 1 and 2, however, showed a normal mucosa and no anatomical abnormalities other than a narrow esophagus and gastroesophageal reflux in patient 2. As NRF2 is involved in several metabolic pathways and constant activation of the stress response will deplete cells of many substrates, poor weight gain could have many reasons. Interestingly, birth measurements in all four patients were also normal indicating that chronic NRF2 activation causes primarily a postnatal disorder. Early diagnosis could therefore facilitate pre-symptomatic treatment. Analysis of amino acids including homocysteine and G6PD activity is included in many newborn screening programs thus diagnosis at a very young age would be possible in many countries.

It has been shown previously that Nrf2 suppresses and thereby controls inflammatory reactions[29]. In Nrf2-deficient mice cigarette smoke leads to more severe lung inflammation and sepsis is associated with significantly increased mortality[30, 31]. Myeloid cell-specific deletion of Keap1 in mice with subsequent enhanced levels of Nrf2 on the other hand, was shown to be protective of mortality in polymicrobial sepsis[32]. Moreover, it was shown that the innate immune cells in this mouse model have preserved or possibly enhanced antibacterial defenses. The fact that all four patients suffered from frequent severe infections of the lung and the skin is therefore surprising. In support of a systemic immunodeficiency we found reduced levels of immunoglobulins, a reduced number of switched memory B-cells and a missing antibody response to pneumococcal vaccine. Treatment with intravenous immunoglobulins (IVIG) in patients 2 and 3 did not reduce the frequency or severity of respiratory infections, possibly due to the preexisting lung and sinus disease present prior to initiation of IVIG therapy and perhaps coincident T-cell dysfunction.

MR imaging in patients 1–3 showed a similar pattern with small and large confluent T2 hyperintense lesions in the supratentorial white matter resembling lesions seen in multiple sclerosis (Fig. 1). Moreover, analysis of CSF in patient 1 (not investigated in patients 2 and 3) revealed oligoclonal bands.

However, no contrast enhancement was seen on MRI and serial follow-up over 3 years in patient 1 did not reveal dissemination in time making the diagnosis of multiple sclerosis unlikely. Quantitative MR studies (patient 1) assessing myelination, namely the parameter map of MT sat, demonstrated markedly diminished myelin content within the lesions indicating brain tissue damage (Fig. 1d). Also supportive of a neurological phenotype associated with chronic NRF2 activation, are the developmental delay and chronic headaches evident in our patient group. Thus far, NRF2 has been regarded a protective factor in the central nervous system[33]. Stroke and traumatic brain injury have been reported to have a worse outcome in Nfe2l2-deficient mice due to increased oxidative stress and disorders like amyotrophic lateral sclerosis, Alzheimer's disease and Parkinson's disease have also been linked to reduced NRF2 levels. Moreover, Nfe2l2-null mice were found to have vacuolar (spongiform) leukoencephalopathy with widespread astrogliosis[34]. It would appear that not only lack of NRF2 but also chronic activation may cause pathology in the CNS, highlighting the importance of the elaborate mechanisms that control NRF2 activity. One can only speculate why there is leukoencephalopathy in NRF2 patients. As a similar pattern of white matter lesions can be seen in mitochondrial disorders, it might be caused by toxic metabolites or an energy deficit caused by the chronic NRF2 activation. While we did find a mild elevation of the mitochondrial marker lactate in blood and CSF it was not elevated in the proton MRS. However, as in many patients with mitochondrial disorders, lactate might be only elevated under stressful conditions, in particular during infections.

Research so far has concentrated on the beneficial effects of NRF2. Many drugs have been developed to enhance NRF2 levels and some like dimethylfumarate (DMF) are in clinical use. While moderately elevated levels of NRF2 seem to have beneficial effects in different mainly autoimmune disorders the clinical picture seen in the patients presented here indicates that chronic high levels of NRF2 have negative effects.

Other drugs have been developed to reduce elevated levels of NRF2 for the control of cancer[23]. A possible approach is upregulation of an alternative KEAP1 independent pathway that downregulates NRF2 via the glycogen synthase kinase-3 (GSK-3). GSK-3 catalyzed phosphorylation forms a phosphodegron in the Neh6 domain of NRF2 thereby recruiting the substrate receptor β-transducin repeat-containing protein (βTrCP) which enables ubiquitination and consecutive degeneration of NRF2[35, 36].

We performed a literature search of all drugs that have been described to lower Nrf2 levels in order to identify candidates for the treatment of the patients described here. Luteolin and ascorbic acid were chosen because they are considered harmless and without side effects when applied in the commonly used dosages[37]. Luteolin (3′,4′,5,7-tetrahydroxyflavone), is a flavone found in leaves that has been shown to downregulate NRF2 levels independent of KEAP1 by enhancing NRF2 mRNA turnover and reducing NRF2 binding to AREs[38, 39]. To test luteolin's suitability in the specific situation of activating NRF2 mutations we firstly analyzed the effect in fibroblasts of patient 1 and found that 24 h treatment indeed resulted in a significant reduction of NRF2 levels leading to downregulation of AKR1B10 and AKR1C1 mRNA (Fig. 4). In a separate experiment the fibroblasts were treated with ascorbic acid but this treatment was less effective to reduce NRF2 levels (Supplementary Fig. 5, Supplementary Table 5). We initiated treatment of patient 1 with 50 mg luteolin and 200 mg ascorbic acid given orally once daily. No side effects were reported and the effect was surprisingly positive. According to the parents after 6 months of treatment the frequency of infections had reduced and the muscle strength and endurance had increased. Moreover, general school performance had improved. However, this is clearly a preliminary result to be viewed with caution as only one patient has received treatment so far.

In conclusion, we present a novel disorder caused by inborn activating mutations in NFE2L2 leading to widespread misregulation of gene expression and an altered cytosolic redox balance. The clinical hallmarks of the disorder are failure to thrive, immunodeficiency and leukoencephalopathy. Due to the unique laboratory findings of hypohomocysteinaemia and elevated G6PD activity early diagnosis within the framework of newborn screening programs would be possible enabling pre-symptomatic diagnosis and early therapeutic intervention. Finally, our findings challenge the very positive picture of NRF2 that currently exists in the literature and indicate that caution should be applied in the use of medications known to lead to NRF2 accumulation.

## Methods

**Quantitative MT imaging**. MT imaging was performed on a 3T clinical MR system (Tim Trio, Siemens Healthcare, Erlangen, Germany) using a 3D FLASH sequence with 1.25 mm isotropic resolution and 240 mm field-of-view. MT contrast was imposed upon a proton density reference by applying a 12.8 ms Gaussian MT-pulse (540° nominal flip angle 2.2 kHz off resonance)[40]. By means of a second T1-weighted reference (TR/α = 11 ms/15°, 1.5 min), maps of the percentage MT sat were calculated. Data processing used the routines of the FSL 4.1 software library of the Centre for Functional Magnetic Resonance Imaging of the Brain (Oxford, UK). The cyan-blue-gray-red-yellow color scale of the MT sat maps covered a range from −0.1 pu (cyan; cerebrospinal fluid) to 1.2 pu (gray, gray matter) to 2.5 pu (yellow; white matter). Myelinated WM of controls (MT sat >2.5 pu) appeared uniformly yellow. Red indicates partial volume of white and gray matter. Dark blue indicates edema or partial volume of cerebrospinal fluid and brain tissue.

**Mendeliome sequencing and bioinformatic analysis**. DNA samples were obtained from four patients and parents following informed consent and approval by the ethic commission from the University Medical Center Göttingen, Göttingen, Germany (patient 1), in patient 2–4 genetic testing was performed as a component of routine clinical care with informed parent consent. 'Mendeliome' gene panel data, which includes 4813 disease associated genes, was generated from the index patient (patient 1) as well as the parents using next-generation sequencing approach in cooperation with Cologne Center for Genomics (CCG, University of Cologne, Germany). Used DNA was extracted from peripheral EDTA blood using standard protocols. For each Mendeliome, 50 ng of DNA was fragmented, barcoded and enriched for the TruSight™ One Sequencing Panel (Illumina, San Diego, CA, USA) using Nextera library preparation technology. Purified and quantified library pool was subsequently sequenced on an Illumina MiSeq sequencing instrument (Illumina, San Diego, CA, USA) using a multiplex paired end 2 × 150 bp protocol with 3 Mendelioms per run. Data processing, analysis and filtering were performed using the 'Varbank' GUI and pipeline version 2.14 (CCG,

University of Cologne, Germany) (https://varbank.ccg.uni-koeln.de/). Reads were mapped to the human genome reference build hg19 using the BWA-aln alignment algorithm. GATK v.1.62 was used to mark duplicated reads, to do a local realignment around short insertions and deletions, to recalibrate the base quality scores and to call SNPs and short Indels. The GATK UnifiedGenotyper variation calls were filtered for high-quality (DP > 15; AF > 0.25; QD > 2; MQ > 40; FS<60; MQRankSum > −12.5; ReadPosRankSum > −8; HaplotypeScore < 13) rare (MAF ≤ 0.005 based on 1000 genomes build 20110521 and EVS build ESP6500 and the Exome Aggregation Consortium (http://exac.broadinstitute.org/)) variants, predicted to modify a protein sequence or to impair splicing, implicated by reduced maximum entropy scores (MaxEntScan). The DeNovoGear software was used to identify de novo mutations. False positive and irrelevant variants were further reduced by taking advantage of the Varbank InHouseDB containing 511 epilepsy exomes. Based on the Trio-patient-parent sequencing approach de novo, compound heterozygous and homozygous/hemizygous variants were extracted for the patient. Prediction of functional impact of all received variants was performed using the dbNSFP version 3.0a36,37. DbNSFP software co-applied several in silico analysis tools to predict the conservation, using PhastCons and GERP, and the functional consequence, using SIFT, PolyPhen2, Provean, LRT, MutationTaster, MutationAssessor, FATHMM, VEST, MetaSVM, and MetaLR, of the affected site. In addition phenotype genotype correlations were investigated using public database Online Mendelian Inheritance (OMIM) (http://www.omim.org/), Orphanet (http://www.orpha.net) and ClinVar (https://www.ncbi.nlm.nih.gov/clinvar).

Whole exome sequencing of patients 2 and 3 was performed at GeneDx. Genomic DNA was extracted from whole blood from affected children and their parents. Whole exome sequencing was performed on exon targets captured using the Clinical Research Exome kit (Agilent Technologies, Santa Clara, CA). (Tanaka et al.—PMID 26299366)[41]. The general assertion criteria for variant classification are publicly available on the GeneDx ClinVar submission page (http://www.ncbi.nlm.nih.gov/clinvar/submitters/26957/).

NFE2L2 mutations were validated by Sanger sequencing (primer information is available in Supplementary Table 6a).

**Cell culture**. Human skin biopsy from the NRF2 p.Thr80Lys mutant patient 1 was obtained at the Georg August University, Department of Pediatrics and Pediatric Neurology, after informed consent. Permission by the ethics committee of the University Medical Center Göttingen, Göttingen, Germany has been granted (Nr. 2/5/16). Patients 2 and 3 refused a skin biopsy.

Human primary skin fibroblasts were extracted from NRF2 p.Thr80Lys mutant human skin biopsy, after informed consent, and maintained as monolayer cultures in Dulbecco's modified Eagle's medium (DMEM/low glucose) supplemented with 10% fetal bovine serum (FBS), 2 mM L-glutamine and 100 U/ml penicillin, 100 µg/ml streptomycin. All reagents were purchased from Biochrom GmbH, Berlin, Germany. Cells were incubated at 37 °C in an atmosphere of 5% CO2. Stock banks were prepared to have early cell passage available.

For molecular biological investigations, NRF2 p.Thr80Lys mutant and wild-type cells were seeded with 1 × 10^6 cells per 10 cm culture plate in a total volume of 10 ml. After 2 days of cultivation all cells were washed with PBS Dulbecco (Biochrom, Berlin, Germany) and harvested using cell scraper. During cultivation, the culture medium was changed after 1 day. Control plates included fibroblasts from healthy patients negative tested for NRF2 variants in the DLG and ETGE motif (NRF2 WT 1—WT 5). All cell cultures were regularly tested with PCR-based test for detection of mycoplasma contamination. In addition, cells were validated for NFE2L2 mutation status using Sanger sequencing (primer information is available in Supplementary Table 6a).

**MG-132 and D,L-sulforaphane treatment**. In order to judge the NRF2 antibody specificity and sensitivity, NRF2 p.Thr80Lys mutant and wild-type cells were treated separately with the proteasome inhibitor MG-132 (10 mM in DMSO, Cell Signaling, Cambridge, UK) and D,L-Sulforaphane SFN (10 mM in DMSO, Sigma, Saint Louis, MO, USA) as recommended. DMSO was used as a control. Cells were treated with 10 µM MG-132 or 10 µM D,L-Sulforaphane for the last 16 h before collecting. Stabilization of NRF2 was analyzed by immunoblotting. The specificity of the used NRF2 antibody was confirmed as advised in the literature[42] (Supplementary Fig. 3).

**Luteolin and ascorbic acid treatment**. NRF2 p.Thr80Lys mutant fibroblast cells of patient 1 were seeded with 3 × 10^5 cells per well in a six-well plate in a total volume of 2 ml. After 1 day, cells were stimulated for 24 h with the flavone luteolin or ascorbic acid. Luteolin (≥98 purity, Sigma-Aldrich) was solubilized in DMSO to obtain a 20 mM stock solution. Ascorbic acid (≥99 purity, Roth) was solubilized in H2O to obtain a 100 mM stock solution. Stimulation of cells was performed in FBS-free DMEM cell culture medium supplemented with a final concentration of 50 µM Luteolin or 0.1 mM, 0.25 mM, 0.5 mM, 0.75 mM, and 1 mM ascorbic acid. Control cells were incubated with the same amount of DMSO (0.25%) or without any supplements. Effect of luteolin or ascorbic acid on the expression of NRF2 was analyzed by immunoblotting. All treatment experiments consisted of at least three independent replicates. Effect of luteolin on the mRNA level of the NRF2 targets AKR1B10 and AKR1C1 was analyzed by qRT–PCR.

**qRT-PCR**. Total RNA was extracted from non-treated NRF2 mutant and wild-type fibroblasts using NucleoSpin RNA Kit (Macherey-Nagel, Düren, Germany) as recommended by the manufacturer. RNA quality was verified by gel electrophoresis and OD measurements. For first-strand cDNA synthesis 2 μg of RNA was reverse transcribed using oligo(dT)$_{15}$ primers and SuperScript III First-Strand Synthesis System (Invitrogen, Karlsruhe, Germany) according to manufacturer's recommendation. qRT-PCR was based on the SYBR green technology using the iQ SYBR Green Supermix kit (BioRad Laboratories, Munich, Germany). The qRT-PCR reactions were performed in triplicates on the MyiQ Single-Color Real-Time PCR Detection System (Bio-Rad Laboratories, Munich, Germany) at annealing temperatures of 60 °C and specificity controlled by post-amplification melting curve analysis. RT-PCR quantification was performed according to the $\Delta\Delta_{CT}$ method (Livak and Schmittgen, 2001)[43]. Data were calculated based on the housekeeping gene *ACTB* and the control fibroblasts NRF2 WT 1. Additional internal reference genes were *GAPDH* and *PGK1*. *NFE2L2* and the interacting partner *KEAP1* as well as described NRF2 target genes *GSR, GCLM, G6PD, ME1, PRDX1, TXNRD1, ABCC1, HMOX1, AKR1B10,* and *AKR1C1* were examined. Primers flanked an intron with amplicon length <150 bp were designed using qPrimerDepot (https://primerdepot.nci.nih.gov/) and ordered by Integrated DNA Technology (IDT, Leuven, Belgium). Only primers with an amplification efficiency of ~ 2 were used. Detailed primer information is available in Supplementary Table 6b. All qRT-PCR experiments consisted of at least three independent replicates.

**Immunoblotting**. Fibroblasts were lysed in SDS-sample buffer (25 mM Tris, 1% SDS, pH 7.5) and rapidly frozen with liquid nitrogen. After quick thawing, lysates were denatured at 95 °C for 5 min. For DNA digestion lysates were incubated at 37 °C for 15 min with benzonase nuclease (Santa Cruz biotechnology, Dallas, TX, USA). Whole protein lysates were clarified by centrifugation (13,000 × rpm, 10 min). Protein concentration was measured using BC Assay protein quantitation kit (Interchim, Montlucon,France) and samples were diluted in 4XLämmlibuffer with DTT (320 mM Tris/HCl, 8% SDS, 20% Gycerol, 0.1% bromphenol blue, 0.6 M DTT, pH 6.8). A total of 25 μg of protein lysates were separated by 10% SDS-Page and transferred onto nitrocellulose membrane using semi dry blotting. Membranes were blocked in 5% nonfat milk in PBS buffer with 0.1 % Tween and then immunoblotted overnight at 4 °C using the following primary antibodies at the specified concentrations: ACTB (AC-15, mouse, Sigma-Aldrich, 1:10,000 dilution), NRF2 (D1Z9C, rabbit, cell signaling technology, 1:1000 dilution), KEAP1 (D1G10, rabbit, Cell Signaling Technology, 1:1000 dilution), and G6PD (D5D2, rabbit, cell signaling technology, 1:1000 dilution), AKR1B10 (ab96417, rabbit, abcam, 1:1000 dilution) and AKR1C1 (ab192785, rabbit, abcam, 1:1000 dilution). Secondary HRP-labeled antibodies were obtained from Jackson ImmunoResearch Laboratory. Immunoblotting detection using Lumi-Light and Lumi-Light Plus blotting substrate (Roche, Mannheim, Germany) was documented with CCD digital imaging ImageQuant LAS-4000 system (GE Health care Life Sciences, Freiburg, Germany). All immunoblot experiments consisted of at least three independent replicates. Quantification of the Immunoblotting data was performed using ImageJ[44]. Data were normalized to the reference gene ACTB and the control fibroblasts NRF2 WT 1.

**Measurement of GSR and G6PD activity**. For quantification of GSR activity, erythrocytes from EDTA blood were washed and hemolyzed, and hemolysates were incubated with GSSG and NADPH at 37 °C for the kinetic UV-test as described[45].

For quantification of glucose-6-phosphate dehydrogenase activity, erythrocytes from EDTA blood were washed and hemolyzed, and hemolysates were incubated with glucose-6-phosphate and NADP at 37 °C for the kinetic UV-test as described[46].

**Optical recordings of cytosolic redox conditions**. Cytosolic redox conditions were monitored optically by taking advantage of the optical redox sensor roGFP1 (reduction oxidation sensitive green fluorescent protein 1)[21, 47]. Fibroblast cultures were transiently transfected (lipofectatime 2000, Invitrogen) with a pEGFP-N1/roGFP1 plasmid vector expressing cytosolic roGFP1. Each culture well was filled with 200 μl transfection solution (OptiMEM, Invitrogen) containing 1% lipofectamine plus 1 μg DNA. Upon incubation for 1 h the transfection solution was exchanged with fresh medium. Sufficient levels of roGFP1 expression were obtained within 3 days post transfection.

Excitation-ratiometric redox imaging was performed with a computer-controlled epifluorescence imaging system, which was assembled from a polychromatic xenon-light source (Polychrome II, Till Photonics), a sensitive CCD camera (Imago QE, PCO Imaging), and an upright fluorescence microscope (Axioscop 1, Zeiss). Transfected fibroblast cultures were placed in a submersion-style chamber (30–32 °C) and excited alternately at 395 nm and 470 nm at frame rates of 0.1 Hz. Fluorescence was recorded with a 63x water immersion objective (Zeiss Apochromat, 1.0 NA), a 492 nm shortpass excitation filter, a 495 nm dichroic mirror, and a 525/50 nm bandpass emission filter. The roGFP1 fluorescence ratio ($F_{395}/F_{470}$) was determined within defined regions of interest by calculating the mean pixel gray values using the TILLvisION control software of the imaging system (version 4.0.1; TILL Photonics)[48]. For quantitative analysis, the roGFP1 ratio was calibrated to saturating oxidizing (5 mM $H_2O_2$) and reducing

responses (10 mM DTT)[21, 48]. Based on these calibration data, the relative oxidation level of roGFP1 ($OxD_{roGFP1}$) was determined[22, 49]:

$$OxD_{roGFP1} = \frac{RoR_{red}}{\frac{F470_{ox}}{F470_{red}}(R_{ox} - R) + (R - R_{red})}$$

The corresponding cytosolic roGFP1 redox potentials ($E_{roGFP1}$) can then be calculated using the Nernst equation and the standard redox potential of roGFP1 ($E^{0'} = -291mV$)[21, 22, 49]:

$$E_{roGFP1} = E^{0'}{}_{roGFP1} - \frac{RT}{2F}\ln\left(\frac{1nOxD_{roGFP1}}{OxD_{roGFP1}}\right)$$

**Statistics**. All statistics were calculated with IBM SPSS Statistics 24. Final graphs were represented as mean percentages ± SEM. One-way analysis of variance was performed with multiple comparison post-hoc-test for qRT-PCR and immunoblotting analysis. Unpaired Welch's t test was performed for luteolin treatment as well as relative oxidation level of roGFP1 and roGFP1 redox potential. $p \leq 0.05$ was considered significant: *$p \leq 0.05$, **$p \leq 0.01$, ***$p \leq 0.001$. Statistical details are included in Supplementary Table 5.

**Data availability**. The data that support the findings of this study are available from the corresponding author upon request.

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

## Acknowledgements

We are grateful to Professor S. James Remington (Institute of Molecular Biology, University of Oregon, Eugene OR USA), for making available to us the plasmids expressing roGFP1 redox-sensitive proteins. The work was supported by the Cluster of Excellence and DFG Research Center Nanoscale Microscopy and Molecular Physiology of the Brain (CNMPB) and the German Research Foundation (DFG Ga354/14-1). The Mendeliome analysis was performed on CHEOPS, a high performance computer cluster of the regional data center (RRZK) of the University of Cologne, funded by the Deutsche Forschungsgemeinschaft (DFG). The participation of JAC in this study was supported in part by the Jeffrey Modell Foundation and the Foundation for Primary Immunodeficiencies. We acknowledge support by the Open Access Publication Funds of the Göttingen University.

## Author contributions

J.A., F.M., and A.B. performed exome sequencing, P.H., S.W., H.T., J.A., P.N., and J.G. analyzed exome data, S.W., F.M., and A.B. performed Sanger sequencing, S.W. and A.W. performed Cell culture, immunoblotting and qRT–PCR, W.N.K.-V. performed measurement of GSR and glucose-6-phosphate dehydrogenase activity, S.W. and M.M. performed optical recordings of cytosolic redox conditions, P.H. and S.D.-K. reviewed the patient scans. S.D.-K. performed quantitative MT imaging, P.H., S.W., and J.H. designed and supervised the project and wrote the manuscript supported by B.H., J.A.C., P.N., and R.S.; J.A.C., R.S., and M.K. identified affected patients and assisted with related clinical and laboratory studies.

## Additional information

**Competing interests:** The authors declare no competing financial interests.

