## [Peer Review File · Nature Communications]

Reviewers' comments:

Reviewer #1 (Remarks to the Author):

The authors are to be congratulated on putting together an interesting set of three patients all with de novo mutations in NRF2, demonstrating an overlap in clinical and biochemical phenotype between these affected individuals and providing supportive functional data. I have no particular concerns or comments, other than perhaps to say that the treatment aspect of the paper is relatively weak considering the length of time of treatment and the 'measures' of possible efficacy provided – do the authors have more data now that could be included in a revision? Otherwise, this is a piece of modern clinical genetics very well done, providing potentially important biological inferences.

Reviewer #2 (Remarks to the Author):

NCOMMS-16-30799-T

This paper by Peter Huppke and Susann Weissbach describes three young male patients with mutations in the NFE2L2 gene that are believed to result in constitutive activation of the antioxidant transcription factor NF-E2 p45 related factor 2 (i.e. NRF2) because the mutations arise within either the ETGE motif or the DLG motif in the Neh2 domain of NRF2 to which the repressor Kelch-like ECH-associated protein 1 (i.e. Keap1) binds. Importantly, Keap1 is an adaptor protein for the Cul3 E3 ubiquitin ligase, and so impairment of its binding to NRF2 (which requires both the ETGE and DLG motifs) results in an increased abundance of the transcription factor and an upregulation in expression of its target genes. The authors report that in patients with gain-of-function mutations in NFE2L2/NRF2 is responsible for failure to thrive, immunodeficiency and neurological symptoms. These clinical manifestations of mutant NFE2L2 are also associated with low homocysteine levels and an increase in glucose-6-phosphate dehydrogenase activity.

The paper is novel and potentially interesting, but it is written in an anecdotal style with a paucity of hard analytical data, which will substantially limit its use to the research community.

Point 1. This paper is written as an extended a clinical case report. Much of the data used to compare the three patients with NFE2L2 mutations are semi-quantitative with an absence of controls. Thus results in Table 1 for patients 1, 2 and 3 contain no reference ranges or statistical significance.

Point 2. Page 2, line 37: It should be noted that the gene encoding NRF2 is called NFE2L2. The gene is not called NRF2, and so the authors should adopt this nomenclature to avoid confusion amongst researchers.

Point 3. Pages 2-3, lines 34-56: In the Introduction section of the paper, the authors should mention that NRF2 is negatively regulated by glycogen synthase kinase-3 (GSK-3) and that this kinase forms a phosphodegron in the Neh6 domain of the transcription factor that is bound by beta-TrCP. This is an important fact that has been completely overlooked because it provides a Keap1-independent mechanism by which NRF2 can be repressed through ubiquitination by Cul1. Indeed, activation of the GSK-3/ beta-TrCP axis is a means by which the upregulation of NRF2 could be corrected in patients with NFE2L2 mutations that cause the transcription factor to evade repression by Keap1.

Point 4. Page 3, line 73: Why is there a 'distinct myelin deficit'? What is the reason for the lesion?

Point 5. Page 3, lines 74 and 75: The data supporting the conclusion that upregulation of Nrf2 results in low homocysteine levels in blood should be presented since this is one of the major conclusions of the paper. What are the units for the homocysteine reference range? The authors comment that the low homocysteine levels may be a consequence of high glutathione levels, but no data are presented. Did they observe a corresponding increase in glutathione?

Point 6. The quantitative analyses of the expression of NRF2-target genes in fibroblasts from patient 1 (Figure 3) does not include similar data from patients 2 and 3 that would allow comparisons between patients.

Point 7. Page 4, lines 92 and 93: Did the parents of patient 2 have normal NFE2L2 genes, encoding NRF2 with both ETGE and DLG intact?

Point 8. Page 5, line 99: What were the numerical numbers of the immunoglobulins in these patients?

Point 9. Pages 6 and 7, lines 132-155 (Figure 3b): The case that upregulation of NRF2 is responsible for the disorders seen in patients 2 and 3 would be more credible if western blots for NRF2 were presented along with expression of its target genes. At present no data are presented, and so the putative link between NRF2 upregulation and the pathology is merely speculative for these two patients.

Point 10. Page 7, lines 144-148: What is the relative expression of AKR1B10, AKR1C1 and NQO1 in primary fibroblasts from patient 1. This is of relevance because AKR1B10 and AKR1C1 have been reported to be excellent markers for NRF2 activity in the human; see MacLeod et al., 2009 Carcinogenesis 30, 1571-80.

Point 11. Page 8, lines 172-177: There are a couple of problems with the experiment using ascorbic acid and luteolin. First, no data are presented to show they diminish NRF2 levels/activity in fibroblasts from the affected individuals. Secondly, how is it envisaged that ascorbic acid and luteolin alter NRF2 activity? In particular, in view of the fact that the ETGE or DLG motifs are disrupted in these patients, presumably ascorbic acid and luteolin inhibit Nrf2 in a Keap1-independent manner. Does this involve the GSK-3 / beta-TrCP axis?

Point 12. Page 9, Figure 3a: What is the molecular mass of NRF2 shown in the western blot obtained from primary fibroblasts from patient 1 in Figure 3a?

Reviewer #3 (Remarks to the Author):

This is an interesting manuscript with novel finding. The authors have made a compelling point that indeed de novo activating mutations of nrf2 is occurring in the 3 patients. They have also shown that activating mutation does indeed activates Nrf2 using the fibroblast from these patients. This is of great significance because it could be the first such finding.

There are three major points that needs to be resolved.

1. How were the authors able to conclude that the pathology of the multisystem disorder is directly related to activating Nrf2 mutation. There are many other splice variants which must be occurring in

the exome seq data from these patients. What if this is only an association which may not be contributing the pathological outcomes.

2. How is hypocysteinemia caused by activating mutation of Nrf2?

3. Clearly there should be more validation for the western and gene expression. The normal sample shown is limited in number.

Reviewers' comments:

Reviewer #2 (Remarks to the Author):

This is an extremely interesting paper as the consequences of de novo mutations in the gene for NRF2 that cause an upregulation in expression of NRF2-target genes have not been described before. It is intriguing because it reflects in part changes that occur in cancer that occur upon somatic mutations in NFE2L2 and KEAP1. Also, the possibility that mutations that activate NRF2 cause reductive stress is gratifying. The paper should be widely read.

One important but small point, is that the authors continue to call the gene that encodes NRF2 as NRF2 (in italics). However, the cancer geneticists and mouse geneticists refer to the gene that encodes NRF2 as NFE2L2 (in italics) or Nfe2l2 (in italics). Please make sure the correct nomenclature is used.

Reviewer #3 (Remarks to the Author):

The authors have satisfactorily addressed all my concerns. This has significantly increased the impact of this novel study.

Thank you very much for considering our article “Activating de novo mutations in NRF2 cause a multisystem disorder” for publication in Nature Communications and thank you to the reviewers for their work and their helpful remarks!

We have addressed all the concerns raised by the reviewers point by point. Furthermore we were able to include another patient in the article who also suffers from this novel disorder. The passages in the text that were accordingly changed are marked in yellow.

Reviewer #1 (Remarks to the Author):

The authors are to be congratulated on putting together an interesting set of three patients all with de novo mutations in NRF2, demonstrating an overlap in clinical and biochemical phenotype between these affected individuals and providing supportive functional data. I have no particular concerns or comments, other than perhaps to say that the treatment aspect of the paper is relatively weak considering the length of time of treatment and the ‘measures’ of possible efficacy provided – do the authors have more data now that could be included in a revision? Otherwise, this is a piece of modern clinical genetics very well done, providing potentially important biological inferences.

- We are aware that the data on treatment response is preliminary. Unfortunately all three patients come from non-affluent families and there are still negotiable health insurance issues for the two patients from the US. Therefore, only the German patient is currently treated with vitamin C and luteolin. But this treatment looks very promising and we were able to include three more months of treatment follow up in this patient. (2nd paragraph page 7)

Reviewer #2 (Remarks to the Author):

This paper by Peter Huppke and Susann Weissbach describes three young male patients with mutations in the NFE2L2 gene that are believed to result in constitutive activation of the antioxidant transcription factor NF-E2 p45 related factor 2 (i.e. NRF2) because the mutations arise within either the ETGE motif or the DLG motif in the Neh2 domain of NRF2 to which the repressor Kelch-like ECH-associated protein 1 (i.e. Keap1) binds. Importantly, Keap1 is an adaptor protein for the Cul3 E3 ubiquitin ligase, and so impairment of its binding to NRF2 (which requires both the ETGE and DLG motifs) results in an increased abundance of the transcription factor and an upregulation in expression of its target genes. The authors report that in patients with gain-of-function mutations in NFE2L2/NRF2 is responsible for failure to thrive, immunodeficiency and neurological symptoms.

These clinical manifestations of mutant NFE2L2 are also associated with low homocysteine levels and an increase in glucose-6-phosphate dehydrogenase activity.

The paper is novel and potentially interesting, but it is written in an anecdotal style with a paucity of hard analytical data, which will substantially limit its use to the research community.

- We adopted the style of the manuscript and added further hard analytical patient data. NRF2 is a key regulator for cellular resistance against oxidative stress and toxicity. In more than 1000 basic science papers the role and mode of action are described as well as secondary biological effects in cancer and inflammation. Here we describe for the first time a monogenetic disease whose primary cause is mutations in NRF2. This human NRF2 model will not only help to understand the disease mechanisms and to design treatment approaches but also reveal various general aspects of the complex biology of NRF2 in health and disease. Thus, identification of this NRF2 disease is of great value for physicians but even more for researcher with a primary interest in NRF2.

Point 1. This paper is written as an extended a clinical case report. Much of the data used to compare the three patients with NFE2L2 mutations are semi-quantitative with an absence of controls. Thus results in Table 1 for patients 1, 2 and 3 contain no reference ranges or statistical significance.

- The laboratory patient data reported in the paper have thoroughly been analyzed. Following the reviewers advice we have included the patient values and the laboratory reference range values in table 1.

Point 2. Page 2, line 37: It should be noted that the gene encoding NRF2 is called NFE2L2. The gene is not called NRF2, and so the authors should adopt this nomenclature to avoid confusion amongst researchers.

- *NRF2* is the gene name alias for *NFE2L2*. We have now included both gene names (page 2, 2nd paragraph) So far, we did not change the gene name in the whole text since the vast majority of articles in this research field use *NRF2* as the gene name. If the editors prefer *NFE2L2* instead of *NRF2*, we can adapt it throughout the article.

Point 3. Pages 2-3, lines 34-56: In the Introduction section of the paper, the authors should mention that NRF2 is negatively regulated by glycogen synthase kinase-3 (GSK-3) and that this kinase forms a phosphodegron in the Neh6 domain of the transcription factor that is bound by beta-TrCP. This is an important fact that has been completely overlooked because it provides a Keap1-independent mechanism by which NRF2 can be repressed through ubiquitination by Cul1. Indeed, activation of the GSK-3/ beta-TrCP axis is a means by which the upregulation of NRF2 could be corrected in patients with NFE2L2 mutations that cause the transcription factor to evade repression by Keap1.

- This is valuable advice. We have included a paragraph in the discussion because we feel that it is most relevant for disease pathomechanisms and furthermore a potential therapeutic approach. (page 11, 3rd paragraph)

Point 4. Page 3, line 73: Why is there a 'distinct myelin deficit'? What is the reason for the lesion?

- The T2 weighted MRI showed hyperintense lesions in the white matter. Such lesions are the hallmark of disorders affecting the myelin sheath. Moreover the quantitative MR studies (MTsat) indicate diminished myelin in these areas. However these findings do not point towards a specific pathophysiology. Various disorders caused by enzyme defects like lysosomal storage disorders or peroxisomal disorders, disorders affecting structural protein or the energy metabolism can cause such lesions. We attempted to learn more about the nature of these lesions by performing MR spectroscopy but only found unspecific findings indication tissue damage. We have included results of the MRS analysis and measurements of lactate in the case report and a paragraph in the discussion. (page 3, 2nd paragraph, page 11, 1st paragraph)

Point 5. Page 3, lines 74 and 75: The data supporting the conclusion that upregulation of Nrf2 results in low homocysteine levels in blood should be presented since this is one of the major conclusions of the paper. What are the units for the homocysteine reference range? The authors comment that the low homocysteine levels may be a consequence of high glutathione levels, but no data are presented. Did they observe a corresponding increase in glutathione?

- The reduced homocysteine levels were at first a very unusual clinical observation. In the diagnostic work-up of patients with unclear neurological disorders we frequently measure homocysteine levels. However, so far we have never come across a patient with levels that low.
- We did not succeed to measure glutathione levels in the fibroblasts from patient 1 with different commercially available essays. Possibly there is a problem with the measurement in human fibroblasts. Nevertheless, we measured cysteine in blood. The also reduced cysteine value in this patient further points towards an increased usage for glutathione production. Moreover we reworded the paragraph on glutathione metabolism in the discussion that was unclear. (page 8, 3rd paragraph)
- Furthermore we included an fourth patient with a mutation in the KEAP1 binding site who also has low homocysteine and cysteine.
- We have included the units for the reference range values. (Table 1)

Point 6. The quantitative analyses of the expression of NRF2-target genes in fibroblasts from patient 1 (Figure 3) does not include similar data from patients 2 and 3 that would allow comparisons between patients.

- We would have liked to examine the fibroblasts from patient 2 and 3 but unfortunately both refused to have a skin biopsy taken. – This information was included in the article. (page 5, 4th paragraph)

Point 7. Page 4, lines 92 and 93: Did the parents of patient 2 have normal NFE2L2 genes, encoding NRF2 with both ETGE and DLG intact?

- The DNA sequences encoding the ETGE and DLG motive of NRF2 in all parents were analyzed and found normal.

Point 8. Page 5, line 99: What were the numerical numbers of the immunoglobulins in these patients?

- The numerical numbers are now included in table 1.

Point 9. Pages 6 and 7, lines 132-155 (Figure 3b): The case that upregulation of NRF2 is responsible for the disorders seen in patients 2 and 3 would be more credible if western blots for NRF2 were presented along with expression of its target genes. At present no data are presented, and so the putative link between NRF2 upregulation and the pathology is merely speculative for these two patients.

- As we mentioned above, we unfortunately do not have fibroblasts from patient 2 and 3. However, not only the clinical phenotype but also the laboratory values are almost identical to those seen in patient 1 and thus make it very likely that all three patients suffer from the same disorder. Furthermore we included another patient with a similar phenotype a novel *NRF2* mutation affecting the binding site with KEAP1.

Point 10. Page 7, lines 144-148: What is the relative expression of AKR1B10, AKR1C1 and NQO1 in primary fibroblasts from patient 1. This is of relevance because AKR1B10 and AKR1C1 have been reported to be excellent markers for NRF2 activity in the human; see MacLeod et al., 2009 Carcinogenesis 30, 1571-80.

- We followed this very valuable advice and analyzed the expression of AKR1B10, AKR1C1 and NQO1 using qRT-PCR and Western blot. For AKR1B10 and AKR1C1 expression we found a dramatic increase in the fibroblasts of patient 1 when compared to healthy controls as described in the article mentioned above (Figure 3a, 3b, S Figure 4a, 4b, S Table 5a, 5b). *NQO1* is not sufficiently expressed in human fibroblasts.

Point 11. Page 8, lines 172-177: There are a couple of problems with the experiment using ascorbic acid and luteolin. First, no data are presented to show they diminish NRF2 levels/activity in fibroblasts from the affected individuals. Secondly, how is it envisaged that ascorbic acid and luteolin alter NRF2 activity? In particular, in view of the fact that the ETGE or DLG motifs are disrupted in these patients, presumably ascorbic acid and luteolin inhibit Nrf2 in a Keap1-independent manner. Does this involve the GSK-3 / beta-TrCP axis?

- Following the reviewers advice we have included the results of the luteolin treatment in the fibroblast of patient 1 that show a significant effect on Nrf2 levels (Fig. 4). We did not include this result initially because it is part of an ongoing graduate student project on the treatment of the Nrf2 patients. Within this project we have started to analyze if the effect is transmitted via GSK-3. So far, however, the results are contradictory to current knowledge as we have repeatedly found a strong increase of the phosphorylated GSK-3 isoforms that are meant to be inactive following treatment with luteolin. We therefore cannot answer the question so far if the downregulation of Nrf2 involves the GSK-3 / beta-TrCP axis.

Point 12. Page 9, Figure 3a: What is the molecular mass of NRF2 shown in the western blot obtained from primary fibroblasts from patient 1 in Figure 3a?

- The molecular mass was included in Fig. 3a.

Reviewer #3 (Remarks to the Author):

This is an interesting manuscript with novel finding. The authors have made a compelling point that indeed de novo activating mutations of NRF2 is occurring in the 3 patients. They have also shown that activating mutation does indeed activates NRF2 using the fibroblast from these patients. This is of great significance because it could be the first such finding.

There are three major points that needs to be resolved.

1. How were the authors able to conclude that the pathology of the multisystem disorder is directly related to activating Nrf2 mutation. There are many other splice variants which must be occurring in the exome seq data from these patients. What if this is only an association which may not be contributing the pathological outcomes.

- Functional NRF2 analyses could only be performed in fibroblasts from patient 1 since we were not able to get fibroblasts from patient 2 and 3 as they refused a skin biopsy. Nevertheless, the clinical phenotype and also the laboratory values are almost identical in these three patients and thus make it very likely that all three suffer from the same disorder. Furthermore, no other candidate genes or shared variants were discovered in the exome sequencing. Therefore several lines of evidence point towards a pathogenicity of the mutations:

1. All three variants affect the very small binding site of Keap1. The probability of finding that alteration in three unrelated patients with a similar and very specific clinical phenotype is extremely low.
2. All three variants are de novo and de novo mutations are rare (see "Prevalence and architecture of de novo mutations in developmental disorders patients" Nature 542, 433–438 (23 February 2017))
3. The biochemical results in all three patients can be explained by the defect in NRF2.
4. No variants with other phenotypes in the *NRF2* gene have been reported in PubMed or are registered in genematcher (To date 2900 researcher have submitted variants in more than 6300 genes to this database).
5. Further supporting the pathogenicity of the NRF2 mutations we have included another patient (patient 4) with a similar phenotype and a novel NRF2 mutation also affecting the binding site with KEAP1. This patient also has similar laboratory results.

2. How is hypocysteinemia caused by activating mutation of Nrf2?

- This point was also made by reviewer 2. The paragraph in the discussion was obviously not sufficient and has been improved. We also added the low cysteine level seen in patient 1 and 4 (not analyzed in patient 2 and 3). (page 8, 3rd paragraph)

3. Clearly there should be more validation for the western and gene expression. The normal sample shown is limited in number.

- Following the reviewers advice we have increased the number of controls to 5. As suggested by reviewer 2 we also included two more NRF2 targets, namely AKR1B10 and AKR1C1. (Supplementary fig. 4)

Answers to the reviewers:

Reviewer #2 (Remarks to the Author):

This is an extremely interesting paper by Peter Huppke and Susann Weissbach in which they describe three young male patients and one female patient with mutations in the NFE2L2 gene, which result in constitutive activation of the antioxidant transcription factor NF-E2 p45 related factor 2 (i.e. NRF2) because the mutations arise within either the ETGE motif or the DLG motif in the Nrf2 ECH homology 2 (i.e. Neh2) domain of NRF2 to which the repressor Kelch-like ECH-associated protein 1 (i.e. KEAP1) binds. Importantly, KEAP1 is an adaptor protein for the Cul3 E3 ubiquitin ligase, and so impairment of its binding to NRF2 (which requires both the ETGE and DLG motifs) results in an increased abundance of the transcription factor and an up-regulation in expression of its target genes. Most provocatively, the authors report that the gain-of-function mutations in NFE2L2 results in failure to thrive, immunodeficiency and neurological symptoms; these clinical manifestations of mutant NFE2L2 are also associated with low homocysteine levels and an increase in glucose-6-phosphate dehydrogenase activity. Moreover, the authors report that in one of the four patients studied, treatment with the flavonoid luteolin and ascorbic acid, which in the case of luteolin inhibits NRF2, can alleviate some of the clinical symptoms. The paper is novel because *de novo* gain-of-function mutations in NFE2L2, resulting in aberrant activation of NRF2, have not been described previously. Interestingly, some parallels have been drawn between the phenotypes of the four patients with *de novo* mutations in NFE2L2 and the characteristics of solid tumours such as those of the lung, bladder, head and neck and liver, in which somatic mutations in NFE2L2 [and also KEAP1] commonly occur. Most tellingly, nothing was previously known about in born errors in NRF2 signalling, and the idea that this represents an example of 'reductive stress' is potentially fascinating.

Scientific questions

Point 1. Page 8, lines 179-186: The authors have shown in Figure 4 that treatment of fibroblasts from Patient 1 with 50 microM luteolin for 24 h markedly decreased NRF2 protein levels. However, Patient 1 was treated with both luteolin and ascorbic acid daily for 6 months. The questions therefore arise as to whether treatment with ascorbic acid could decrease NRF2 protein levels in fibroblasts for Patient 1, and whether treatment with luteolin plus ascorbic acid elicited a different effect on NRF2 protein than just luteolin alone.

- We treated the fibroblasts with ascorbic acid but failed to show any effect. The results are included in figure 4.

Point 2. Page 8, line 179-181: Was the decrease in NRF2 protein caused by treatment with luteolin accompanied by a decrease in mRNA for AKR1C1 other NRF2-targets?

- AKR1C1 mRNA was downregulated. Results of this experiment are included in figure 4.

Point 3. Page 12, lines 285-288 and Page 13, lines 314-316: Permanent and constitutive activation of NRF2 resulting from genetic *de novo* mutations is not equivalent to the temporary and pulsatile pharmacological activation of NRF2 achieved by agents such as dimethylfumarate or sulforaphane. Thus it is somewhat disingenuous to suggest that the two are equivalent and that small molecules

that activate NRF2 are potentially harmful. A more nuanced discussion should be supplied by the authors!

- The discussion was altered according to the reviewers comment. Page 13 top

Editorial points

Point 4. Page 1, line 1: The gene encoding the NRF2 protein is called *NFE2L2*, and this is the nomenclature used by The Cancer Atlas Genome (TCAG) consortium in numerous papers (many in Nature) describing their studies of somatic mutations. Indeed, all geneticists refer to the gene encoding NRF2 as *NFE2L2*; they never refer to the gene as NRF2. Thus, the authors of the present paper should adopt this nomenclature to avoid confusion and to ensure their paper is widely read by geneticists. With this in mind, the title of the article should be changed to “Activating *de novo* mutations in *NFE2L2*, which encodes NRF2, cause a multisystem disorder”.

- The nomenclature was altered throughout the text.

Point 5. Page 5, lines 90-91: Remove the sentence- “The correct but rarely used nomenclature for this gene is *NFE2L2*.”

- The text was altered according to the reviewer’s suggestions.

Point 6. Page 3, line 33: Please reword first sentence as- “Transcription factor NRF2, encoded by *NFE2L2*, is a master regulator of defense against oxidative stress in mammalian cells”.

- The text was altered according to the reviewer’s suggestions.

Point 7. Page 3, line 34: Reword as- “In cancer, somatic mutations in *NFE2L2* that lead to accumulation of NRF2 protein promote tumour cell survival and drug resistance.”

- The text was altered according to the reviewer’s suggestions.

Point 8. Page 3, lines 51-59: Having defined the abbreviation for KEAP1 in line 50 in capital letters, the authors subsequently use “Keap1” throughout the rest of the paragraph. Please be consistent throughout m/s.

- The text was altered according to the reviewer’s suggestions. KEAP is used for human cells and Keap1 for animal cell lines.

Point 9. Page 3, line 54: Define the Neh2 domain (it is Nrf2 ECH homology 2, see above).

- The text was altered according to the reviewer’s suggestions.

Point 10. Page 3, line 55: The NRF2 transcription factor contains seven domains

with defined functions. Please see Wang et al. Cancer Research 2013; 73, 3097-3108.

- The text was altered according to the reviewer's suggestions.

Point 11. Page 4, line 61: The article now describes four patients, not three.

- The text was altered according to the reviewer's suggestions.

Point 12. Page 4, line 72: Add a hyphen between "6" and "year", giving "6-year old".

- The text was altered according to the reviewer's suggestions.

Point 13. Page 5, line 94: Do not put "NRF2" in italics because it is the protein that is being referred to.

- The text was altered according to the reviewer's suggestions.

Point 14. Page 5, line 97: Change to "13-year old".

- The text was altered according to the reviewer's suggestions.

Point 15. Page 5, line 101, and Page 6, line 112: Change "Tab. 1" to "Table 1".

- The text was altered according to the reviewer's suggestions.

Point 16. Page 5, line 114: Change to "20-month old".

- The text was altered according to the reviewer's suggestions.

Point 17. Page 6, line 119: Change "data is" to "data are".

- The text was altered according to the reviewer's suggestions.

Point 18. Page 6, line 122: Change "motive" to "motif".

- The text was altered according to the reviewer's suggestions.

Point 19. Page 6, line 125: Change "NEH2" to "Neh2".

- The text was altered according to the reviewer's suggestions.

Point 20. Page 7, line 138: Do not put "NRF2" in italics because it is the protein that is being referred to.

- The text was altered according to the reviewer's suggestions.

Point 21. Page 7, lines 144-146: The abbreviations GCLM, GSR, G6PD, ME1,

TXNRD1, PRDX1, HMOX1 and ABCC1 should probably all be defined at their first

use in the text. Some of these are defined later in the text, and glutathione reductase variously appears in the text in full (e.g. lines 151, 153, 209, 210) or as GSR! It is all a bit unsystematic at present. Please tighten up!

- The text was altered according to the reviewer's suggestions.

Point 22. Page 7, line 147: To accommodate abbreviations suggest rewording as "The strongest increase in expression was seen for the aldo-keto reductase (AKR) 1C1 and AKR1B10 genes..."

- The text was altered according to the reviewer's suggestions.

Point 23. Page 8, lines 169 and 172: Please don't use an apostrophe. Suggest rewording as "whereas for the fibroblasts of patient 1..."

- The text was altered according to the reviewer's suggestions.

Point 24. Page 9, lines 189-190: Reword as "...caused by mutations in NFE2L2 that impair repression of NRF2 by KEAP1."

- The text was altered according to the reviewer's suggestions.

Point 25. Page 9, line 193: Reword as "ETGE motifs in the..."

- The text was altered according to the reviewer's suggestions.

Point 26. Page 9, line 197: Reword as "mutation within the ETGE..."

- The text was altered according to the reviewer's suggestions.

Point 27. Page 9, line 204: Change to "increase in expression", also remove the words "aldo-keto reductase" since AKR should be defined in line 147.

- The text was altered according to the reviewer's suggestions.

Point 28. Page 9, line 208: Reword as "the patients described herein enabled us to analyse the in vivo effects of NRF2 upregulation."

- The text was altered according to the reviewer's suggestions.

Point 29. Page 10, line 218: Add the word "cells" after "HepG2".

- The text was altered according to the reviewer's suggestions.

Point 30. Page 10, line 222: Remove hyphens around "We found".

- The text was altered according to the reviewer's suggestions.

Point 31. Page 11, line 263: Remove hyphen/hieroglyphic immediately adjacent to “large”.

- The text was altered according to the reviewer’s suggestions.

Point 32. Page 12, line 274: Change to “Alzheimer’s disease and Parkinson’s disease”.

- The text was altered according to the reviewer’s suggestions.

Point 33. Page 14, line 332: Should this not be four patients, rather than 3?

- The text was altered according to the reviewer’s suggestions.

Point 34. Page 15, line 360: Change to “patients”.

- The text was altered according to the reviewer’s suggestions.

Point 35. Page 16, line 370: Do not put “NRF2” in italics because it is the protein that is being referred to.

- The text was altered according to the reviewer’s suggestions.

Point 36. Page 17, line 415 and Page 19, line 442: “Data were...”

- The text was altered according to the reviewer’s suggestions.